# Motivation to Donate, Job Crafting, and Organizational Citizenship Behavior in Blood Collection Volunteers in Non-Profit Organizations

**DOI:** 10.3390/ijerph17030934

**Published:** 2020-02-03

**Authors:** Marcello Nonnis, Davide Massidda, Claudio Cabiddu, Stefania Cuccu, Maria Luisa Pedditzi, Claudio Giovanni Cortese

**Affiliations:** 1Department of Pedagogy, Psychology, Philosophy, University of Cagliari, 09123 Cagliari, Italy; davide.massidda@gmail.com (D.M.); dott.claudiocabiddu@pec.it (C.C.); cuccustefania@gmail.com (S.C.); pedditzi@unica.it (M.L.P.); 2Department of Psychology, University of Turin, 10124 Turin, Italy; claudio.cortese@unito.it

**Keywords:** motivation to blood donation, organizational citizenship behavior, job crafting, blood collection volunteers

## Abstract

This study assesses the levels of and relationships between the Motivation to donate, Job crafting propensity, and the Organizational citizenship behavior of blood collection volunteers in a non-profit association. An Italian sample of AVIS (the Italian Association of Voluntary Blood donors) blood donors (N = 1215) actively involved in organizing blood collection, were asked to complete the Italian version of the Volunteer Function Index, the Job crafting scale, and the Organizational citizenship behavior scale. The tools were verified by Confirmatory factor analysis and their relationships were explored using Structural equation modeling for hidden variables. The three constructs have overall high scores. Motivation to donate and Job crafting show a clear correlation, with the latter influencing volunteer Organizational citizenship behavior. The study highlights the need to take into consideration the Motivation to donate, Job crafting and Organizational citizenship behavior of volunteers, particularly in countries such as Italy, where blood collection is almost exclusively carried out thanks to spontaneous, altruistic, and disinterested commitment.

## 1. Introduction

The field of organizational systems for blood collection is a complex and multifaceted one. Since 1989, a European Union directive has guaranteed the supply of whole blood and plasma from unpaid volunteer donors [1]. Generally speaking, there are currently three main blood collection systems in Europe [2,3]. The first group consists of countries such as the United Kingdom, France, and Ireland, where blood collection is assigned to the national health service. In the second group, in countries such as Belgium, Luxembourg, and the Netherlands, the Red Cross has a supply monopoly. In Germany, the Red Cross has a majority share in the management of blood collection.

The third group comprises countries such as Greece, Italy, Norway, Portugal, and Spain, where the majority of blood collection is carried out by associations, with a minority share (variable from one country to another) of collection conducted by the Red Cross. In Denmark, France, Greece, Spain, and Italy, blood donors are organized in one or more national associations.

In the particular case of Italy, most of the blood banks are managed by AVIS (the Italian Association of Voluntary Blood donors). With 3400 offices throughout the country, over 1,300,000 blood donors, and about 2,000,000 bags of blood collected per year, AVIS is able to meet about 70% of national needs [4]. As well as managing and organizing volunteers who are dedicated to the collection of blood, the association is committed to supporting and strengthening donor motivation, as well as ensuring that blood is a voluntary gift destined for an unknown person and is non-binding in terms of any expected recompense [5,6].

The already complex and multifaceted scenario of blood collection in Europe was further compounded by the global economic crisis in the early 2000s, whose effects on welfare and on those societies most affected by economic woes are still being felt. A number of studies (see for example Mucci et al. [7]) have highlighted how the current economic crisis has had evident repercussions on the physical and mental well-being of the working populations of several countries around the world. Furthermore, in some particularly vulnerable southern European countries such as Greece, Sotiropoulos et al. [8] have described the manner and process by which citizens and voluntary associations have come together to combat the effects of the economic, social, and health welfare crises in order to compensate for a disintegrating state welfare system.

In European countries like Italy, where blood collection is managed almost exclusively by non-profit organizations, it is of the utmost importance that volunteers are available not only to continue to donate blood, but also to become more involved and actively engaged in a systematic and effective manner in the activities of organizations that promote its collection. Such commitment requires an effort to adapt and innovate the organizational processes and strategies in consideration of the social and cultural changes affecting the donor community (actual or potential), and to oversee the relationship with the many stakeholders involved in the collection and distribution of blood. These include the health system as a whole (hospitals, sorting facilities and blood treatment, patients receiving blood and its derivatives), and the local organizations that are able to facilitate its collection, such as schools or other relevant bodies [3]. Competence and a commitment of this nature becomes essential when, in situations of crisis such as in the aforementioned current one, the state is unable to provide effectively for the health of its citizens, in particular for the collection and distribution of a primary commodity such as blood.

Scientific studies that have focused on organizational/psychological constructs relating to volunteers involved in non-profit blood collection associations still remain very scant. For example, Bani et al. [2] conducted a study on organizational commitment and its relationship with different motivations to volunteering, on a sample of Italian donors (N = 871). The authors have shown how important this construct is for the purposes of the association.

In a study conducted with members of various voluntary associations (N = 1445), including some for blood donation, Lo Presti [9] highlighted that various features (for example social and task support, availability of information and appreciation) affect volunteers’ organizational commitment, job satisfaction, and their intention to remain involved. In another study conducted with a sample composed of non-profit organizations for fundraising members (N = 314) and blood (N = 298), Boenigk et al. [10] highlighted how organizational identification and identity salience together affect satisfaction, loyalty, and behavior; they proposed a series of strategies to implement donor identification management, such as group activities, and development of online communities.

In a study conducted on a sample (N = 2464) of blood donors, Pozzi et al. [11] highlighted how levels of satisfaction with the organization influences wellness and gratification according to the level of social integration, especially regarding the general inclination to continue to be volunteers and donors.

For European countries like Italy (and some other countries), where the collection of blood is almost exclusively assigned to non-profit voluntary organizations, it is therefore of fundamental importance to consider not merely the motivation to donate, but also the propensity to nurture improvement and innovation in carrying out activities and consolidating voluntary commitment to organizational activities in general.

What has been argued so far highlights the importance of motivation in blood donation, the inclination to innovate and improve voluntary commitment to these associations’ organizational activities, with particular attention to discretionary and non-prescribed ones. These last two constructs can be defined respectively as: a propensity for Job crafting (JC) [12,13,14] and Organizational citizenship behaviors (OCB) [15,16].

### 1.1. Motivation for Blood Donation

Given the importance of blood donation as a topic, it has long been studied within different disciplines in the fields of healthcare, social welfare, and economics [17,18]. As a gesture of solidarity and altruism, blood donation is indicative of the importance of reciprocity and exchange between individuals, something which goes beyond the utilitarian logic of the free market [17,19,20]. Moreover, due to its ethical and cultural significance, it cannot be equated with the individualistic and self-relating logics of society today [21].

Psychology has long been interested in the motivations inspiring people to donate blood, from various theoretical perspectives [22,23,24,25,26,27,28].

What has been particularly interesting and useful for this research is the functionalist approach proposed by Omoto and colleagues [29,30,31,32], which states that volunteering fulfills different functions for some individuals, but which may differ notably for others.

According to Omoto et al., people are driven to act by a multiplicity of motivations, which will change over time and in relation to the context in which they are located. Motivation may be simultaneous and changeable over time (e.g., depending on people’s life cycle phase), but it is necessary to consider them together in order for them to be fully understood. The functionalist approach has identified a wide range of motivations. Re-examining Omoto and his colleagues work, Clary et al. [33,34] have identified the following six factors that induce motivation:

- Values, meaning the possibility of finding a context in which the volunteer can express and share his altruistic and humanitarian values.

- Understanding, which refers to the possibility of acquiring new knowledge, skills, and abilities.

- Socializing, which involves the opportunity to meet new people or to carry out an activity with a friend.

- Career advancement, which concerns the potential to acquire and practice useful skills for their profession and to discover new job opportunities.

- Ego-protection, which addresses the defense of the ego, with particular reference to the fact that people often feel guilty for being more fortunate than others. Therefore, by showing commitment to social issues, they can obviate such feelings.

- Self-enhancement, which refers to how engaging in volunteering improves mood and augments self-esteem, a sense of self-worth, and in general develops the concept of the self.

### 1.2. Organizational Citizenship Behavior

Organ [15] has defined organizational citizenship as the set of discretionary behaviors manifested by members of an organization, which favor overall efficiency and effectiveness, even if they are not bound by a contract or explicitly recognized and codified in a system of formal rewards. Falvo et al. [35] describe them as spontaneous, pro-social behaviors that go beyond the prescribed role and can therefore be identified as extra-role behaviors.

The definition of the five dimensions of OCB proposed by Organ is as follows:

- Conscientiousness, i.e., behaviors indicating that a worker is mindful when carrying out his/her work (for example keeping scrupulously to working hours).

- Civic virtue, i.e., behaviors showing a high sense of responsibility towards the organization (for example, by endeavoring to solve problems or offering suggestions for their solution).

- Sportsmanship, i.e., manifesting loyalty to the company, focusing on its best aspects and avoiding singling out less positive ones.

- Altruism, i.e., behaviors expressing availability to help colleagues (for example, helping those who have a greater workload).

- Courtesy, i.e., actions that foster relationships characterized by kindness and co-operation (for example trying to avoid conflict).

Organ’s work was a catalyst for the production of a large number of studies which have analyzed the nature, identified the antecedents, and described the consequences of OCB (i.e., references [36,37,38,39,40].

One of the difficulties we faced with regard to the specific objectives of our study, was the lack of literature on OCB in voluntary organizations. One example, however, is a study of a sample of volunteer Swiss nationals (N = 2222, mainly operating in “Caritas” and the Red Cross), in which Van Schie et al. [41] showed how OCBs that are intended to benefit the organization as a whole (OCBOs) are one of the positive outcomes of motivation in volunteers. They also demonstrated how such motivation is determined by the following three factors: motivational potential of the tasks that volunteers carry out; the encouragement from supervisors for them to take personal initiative; and the correspondence between their own values and those of the company. In another longitudinal study conducted on a sample of Spanish social-work volunteers (N = 419), Aranda et al. [42] showed how a breakdown in harmony and understanding between the association and the volunteers and the negative climate that it causes, will in time undermine their OCB towards both the organization and other volunteer colleagues.

### 1.3. Job Crafting

In recent decades, changes in the labor market have forced companies to improve their skills and knowhow in order to become more competitive. These systematic, rapid (and sometimes tumultuous) changes have directly and indirectly involved workers and organizations [43,44], and have encouraged greater flexibility and stronger personal initiative. It has therefore become progressively more important to develop and improve new strategies that facilitate the success of individuals in coping with labor contexts that are subject to change [45].

Many studies (i.e., reference [46]) have shown that workers who take a proactive role in reshaping their work activities (i.e., job-crafting) are able to bring about desired changes. According to Tims et al. [12,13], JC implies self-initiated behaviors that employees adopt in order to harmonize their work with their preferences, motivations, and needs.

Tims et al. [14] incorporate JC into the Job Demands-Resources Model (JD-R), a theoretical framework which considers two broad categories of dimensions (job demands and job resources) pertaining to well-being and/or work-related stress [47]. In this perspective, JC is defined as the actions that employees may embrace with a view to balancing their job demands and job resources with their own personal abilities and needs [12,13].

Cenciotti et al. [48] recently undertook a study that focused on three positive dimensions of Job crafting, based around the notion of “increasing”:

- increasing structural job resources, which advances people’s inclination to develop their own abilities to learn new skills for personal development;

- increasing social job resources, which refers to furthering collaboration and actively seeking feedback on one’s own work from colleagues, as well as being guided and directed by one’s superiors;

- increasing challenging job demands, which implies the inclination to independently seek, propose, or willingly accept involvement in new activities, projects, and challenges, even when this is not required [14,48].

Therefore, JC is an individual’s strategy to promote change and innovation as well as to embrace challenges that can generate an overall improvement in work organizational processes.

There is very little current scientific literature on the JC construct in non-profit organizations, particularly with reference to voluntary work. However, a study by Millette et al. [49] in Canada, on a sample of volunteers and their supervisors in the field of social work (N = 124), did find indirect evidence that the kind of tasks undertaken by volunteers in the association and the motivation to take personal initiative and find satisfaction therein, do influence both performance and OCB.

Based on what we have argued so far, we believe that the inclination of volunteers to actively engage in increasing the resources available to their association (which can be defined as JC in the voluntary sector), is a critical factor in achieving the objectives of non-profit organizations and thus deserves further investigation and study.

### 1.4. The Relationship Between Blood Donor Motivation, Job Crafting, and Organizational Citizenship Behavior

As we have stated above, there is a growing amount of literature on the three constructs we consider. With regard to the relationship between them, although not carried out in the context of voluntary work, some recent research has confirmed the connection between Job Crafting and OCB. For example, in a study conducted among hospitality workers in India using the above-cited JD-R Model [47], Srivastava et al. [50] showed that the extent of JC in terms of one’s ability to manage one’s own resources and seek new challenges, does reinforce OCB. In a longitudinal study conducted on a sample (N = 50) of Korean flight attendants, again applying the JD-R Model, Shin et al. [51] showed that the amount of daily JC, increasing job resources, and daily challenge job demands have the effect of reinforcing daily OCB. The ability of JC to positively influence OCB was also identified by Guan et al. [52] in their analysis of relations between bosses and workers in a Chinese manufacturing context (N = 406).

However, we have found no noteworthy literature on the topic of the relationship between JC and blood donor motivation, or between donor motivation and OCB, in spite of the significance that these constructs have been shown to possess (in particular in countries like Italy, where blood collection organizations are almost entirely in the voluntary sector).

### 1.5. Theoretical Framework

Our study rests, in general terms, on the theoretical underpinnings of the JD-R Model [47]. In fact, Tims et al. [13,14] have defined the JC construct as being an individual job resource.

Secondly, although the authors themselves (Omoto et al. [29,30,31]) did not explicitly refer to the concept of donor motivation, it is a positive motivational force that should be categorized as a resource.

Further, OCBs were indicated by the authors of the JD-R Model themselves [53] as having a positive organizational effect on well-being and commitment, and may be ascribed to the category of extra-role performances.

Lastly, Demerouti et al. [54] also employed the JD-R Model to corroborate that JC is a construct capable of fostering extra-role positive behaviors, which as we have said can be filed under OCBs (see also references [50,51]).

In focusing our study within the boundaries set by the JD-R Model, we were somewhat limited by the lack of empirical research on relations between JC, donor motivation and OCBs. For this reason, we were unable to formulate robust hypotheses concerning relations between these three constructs.

### 1.6. Objectives and Hypotheses of the Present Study

The present descriptive and exploratory study focuses exclusively on blood donors in non-profit blood collection associations, who have an active, structured, and systematic role in their organization’s activities.

In particular, our aim has been to describe their motivation for blood donation, their propensity to JC (construct of inclination towards improvement and innovation) and OCB (construct of commitment in non-prescribed and extra-role activities) with a view to promoting the good functioning of the association.

For this reason, our primary objective was to carry out a psychometric evaluation with regard to adapting the JC Scale [13,14] and the OCB Scale [16] to the Italian voluntary work context. In addition, we intended to confirm the validity of the Italian version of the instrument measuring donor motivation (Voluntary Function Index [29,30,31]).

Our second objective was to determine a description of the sample of voluntary donors working in a blood collection organization in terms of age and social status variables, etc.

Lastly, and as previously stated, since there is a lack of studies on the relationships between these constructs within the theoretical framework of JD-R Model, we assume that motivation for blood donation underpins the propensity to involve volunteers, and JC predicates the propensity to adapt, change, and improve organizational activities. Our third objective was therefore to assess the influence of these two psychological constructs (job resources) on the volunteer’s OCB, as this particular construct can be viewed as a positive behavioral outcome which manifests in these types of associations.

## 2. Materials and Methods

### 2.1. Sample and Procedure

The research was cross-sectional and conducted via an online research protocol consisting of self-reporting questionnaires. In order to simplify the completing of the protocol, for all the questionnaires, a 5-step response Likert scale was standardized as follows: from 1 (“Not at all true”), to 5 (“Totally true”). The researchers received only valid and fully completed protocols. Data was collected between February 2018 and January 2019. The protocol was sent for completion only to unpaid voluntary donors involved in carrying out the association’s activities.

The sample (N = 1215) consisted exclusively of blood donors who play an active, systematic and continuous voluntary role in the organizational and management activities of Italy’s main blood collection association (AVIS). The sample had an average age of 47.81 years (SD = 14.48), with more than half (55.1%) coming from northern Italy, 25.97% from the center, and 18.93% from the south of the Italian peninsula.

About two thirds of the sample were male (65.02%), with more than half (56.30%) having over 11 years of experience with the association (22.22% had below 5 years and 21.48% between 6 and 10 years); most of the volunteers held a diploma (56.46%) or were university graduates (31.19%). The majority of the sample were employees of some kind (69.71%). Over two thirds of the sample had a managerial role in the association (President, Vice President, Secretary, Treasurer, 69.38%), the remaining 30.62% had executive roles (Executive Committee Member, Auditor, Counselor, ordinary Member).

The sample consisted of individuals who completed the protocol indicating their specific contribution. The sample can be considered homogeneous and representative of all AVIS members active in the organization throughout the country (totaling approximately 17,000 members, who are all unpaid voluntary workers [4]).

### 2.2. Measures

The research protocol was divided into the following sections: (a) instructions on the correct mode of completion in compliance with the Italian legislative Decree on the privacy and anonymity of respondents; (b) a form for the collection of socio-demographic data relating to gender and age, length of service in the association, qualifications, and work situation. Also considered were the role within the association (director/decision-maker vs. executive/operative) and the geographical origin area (north, center, or south Italy); (c) the questionnaires related to the psychological constructs described below.

Motivation for blood donation: Volunteer Function Inventory (VFI). This questionnaire, originally proposed by Omoto et al. [29,30,31] and revised by Clary and colleagues [33,34], was recently adapted for blood donation by Alfieri et al. [55,56,57] for the Italian context. It consisted of 30 items investigating the following six motivational dimensions (five items for each): Values (e.g., “By donating blood, I can do something for a cause that I consider important”); Understanding (e.g., “By giving blood I can learn a lot about the cause for which I am providing my services”); Social (e.g., “I’m worried about who is less fortunate than me”); Career (e.g., “By giving blood I can establish new contacts that may be useful for my job or my career”); Ego-protection (e.g., “Donating blood helps me overcome my guilt for being more fortunate than others “); Self-enhancement (e.g., “Donation makes me feel important”).

Job crafting: the Italian version of the JC Scale by Tims and colleagues [13,14], proposed by Cenciotti et al. [48], was adapted ad hoc to the context of the voluntary association. The tool consists of 15 items that measure the following three dimensions (five items for each): Increasing structural job resources (e.g., “I create the conditions to grow in the association”); Increasing social job resources, (e.g., “I ask advice from my colleagues in the association”); Increasing challenging job demands (e.g., “When there are not very demanding activities, I take the opportunity to propose new initiatives or projects”).

Organizational citizenship behavior: the Italian version of the OCB Scale devised by Podsakoff et al. [16] proposed by Argentero et al. [58], was adapted ad hoc to the context of the voluntary association. The tool consisted of 24 items that measured the following five dimensions: Conscientiousness (five items), e.g., “I respect the regulations of the association even when nobody is observing me”); Civic virtue (four items, e.g., “I participate in activities that are not required but which are important for the image of the association”); Sportsmanship (five items, e.g., “I always look on the positive side rather than the negative side of things “); Altruism (five items, e.g., “ I help anyone who has a lot of work to do in the association”); Courtesy (five items, e.g., “I work to avoid conflict with my colleagues”).

In order to adapt the JC Scale and the OCB Scale to the context of a voluntary association, the terms “organization” and “work” were replaced with the corresponding and more appropriate terms “association” and “activity”, without altering the meaning and the structure of the affected items. These changes were made because in Italian, the terms “lavoro” e “organizzazione” are generally used to refer to habitually paid work and were therefore not congruent with the voluntary, unpaid work activities examined here.

### 2.3. Data Analyses

After inverting the scores of the responses to the items with OCB Scale negative polarity, the consistency of the instruments used was the object of initial scrutiny (VFI, OCB Scale and JC Scale).

In order to check for the goodness of the factorial structure, a Confirmatory factorial analysis (CFA) was performed for each instrument. The models were adapted with maximum likelihood estimation with significant standard errors. The goodness of the adaptation was evaluated jointly considering indices of different types [59]. The indices in question were associated to the classic χ^2^: root mean square error of approximation (RMSEA), a comparative fit index (CFI), a non-normed fit index (NNFI), a comparative fit index (CFI), and a standardized root mean square residual (SRMR), considering the acceptability thresholds described in Hooper et al. [59]. A sufficient adaptation is given by an RMSEA index below 0.07, a CFI index, and an NNFI index of at least 0.95 and an SRMR index lower than 0.08.

In order to define the sample with reference to the psychological variables considered, the total scores of the scales that make up each instrument were calculated for each subject. The calculation was made by considering the average of answers obtained. Given that calibration and reference standards for the Italian context are not available for the questionnaires used [48,55,58], in order to evaluate the value of the sample in question for each construct (and relative dimensions) considered, we decided to use the previously described range of the 5-step Likert response scale as a reference criterion and to consider scores below the intermediate value of the scale (i.e., 3) as low and those above it as high.

In order to verify whether or not the three constructs that relate to the psychological variables identified through the Structural equation model (SEM) vary according to the socio-demographic variables of gender, service age and work role (Managerial vs. Executive), a multivariate variance Analysis (MANOVA) was carried out. This analysis was further investigated using the Tukey post-hoc test.

The second phase involved constructing a SEM in order to compare the measurements. A second-order factor was incorporated for each of the instruments and was considered as an antecedent of the measurements produced by the instrument, thus obtaining three latent macro-factors (each corresponding to the three constructs considered). These three latent variables were placed in relation to one another, assuming that Motivation for blood donation and JC wield an influence on OCB in our context of voluntary associations.

The structure of the model was determined by a selection procedure based on sequence comparisons. Starting from a null model in which the three macro-factors were each defined as being independent structures, the parameters that linked the endogenous variables were gradually released in an attempt to improve data adaptation. Comparison between the models was carried out by means of Akaike Information Criterion [60] and Bayesian Information Criterion (BIC) [61]. The method used to compare the models was to take the difference between the index of a comparison model and the index of a baseline model, thereby obtaining a ΔAIC value and a ΔBIC value.

The analyses were carried out using the statistical environment R version 3.4.3 [62], associated with the lavaan package version 0.5-23.1097 for the CFA [63].

### 2.4. Ethical Issues

The research was authorized by AVIS’s National Governing Body (approval number 18/00239, dated January 10, 2018) and by the Ethics Committee of the University of Cagliari (approval number 0171803, dated July 16, 2019), and was thus conducted in full compliance with the Ethical Principles of Psychologists and Code of Conduct of the APA (American Psychological Association), integrated into the AIP’s (“Associazione Italiana Psicologia”) code of Ethics. The project did not address any sensitive topics and was carried out via online self-reporting procedures for informed and consenting adults. Lastly, in accordance with Italian privacy law, the project ensured the anonymity and privacy of all participants.

## 3. Results

### 3.1. Factorial Analysis of the Instruments

The measurements produced by the instruments generally show an adequate internal consistency, quantified by Cronbach’s alpha index (Table 1).

Exceptions are found in the scales of Social VFI (α = 0.63), in Values (α = 0.50), and in Sportsmanship (α = 0.55) and Courtesy (α = 0.66) of the OCB Scale. Indeed, these scales include items that in some cases have small correlation indices.

The CFA show that all three instruments have contrasting adaptation indices (Table 2). The χ^2^ index is always significant, as would be expected given the high sample size. The RMSEA is within the acceptability threshold for the VFI questionnaire, while it is notably above the threshold for the other two instruments.

The SRMR value always lies within the acceptability threshold, but CFI and NNFI remain far from values considered sufficient for a good adaptation. CFI and NNFI are comparative indices, which evaluate the adaptation of the model in relation to a baseline in which the observed variables are considered to be unrelated; therefore, the relationships between the observed variables may not be high enough for the model architecture to be considered satisfactory.

Calculations were performed on the correlation between estimated factor scores and factor scores calculated as the mean of the observed responses, for all scales of all the instruments considered. These correlations are always particularly high (between 0.88 and 0.91 for VFI, between 0.77 and 0.97 for the OCB scale, and between 0.91 and 0.99 for the JC Scale), making it possible, in practice, to calculate the factor scores directly as the average or sum of the raw scores, without any substantial loss of information. Considering that the capacity of the tools to represent their psychological constructs is consistently maintained, it is possible to state that the VFI, the JC Scale, and the OCB Scale display an adequate factorial structure.

### 3.2. Sample Profiling

The main descriptive statistics of the scores relating to the constructs considered (and their relative dimensions) are shown in Figure 1, while in Table 1 their scores are reported in detail, according to the 5-step Likert scale.

From a largely descriptive point of view, the sample had high enough levels; in fact almost all dimensions registered scores above 3 (the intermediate point of the scale).

Regarding the Motivation to donate (VFI), the highest score was registered in the Values dimension, which denotes a strong tendency of volunteers to consider blood donation as an existential value. Below that, Self-enhancement, Understanding, and Social motivations had very similar values.

The last two motivational factors in donation, Ego-protection and Career, had lower scores (below the intermediate point of the scale).

In regards to JC, and taking all the constituent dimensions, the sample displayed values beyond the intermediate point of the reference scale. The highest scores were registered in the Increasing structural resources dimension, followed by the Increasing challenging job demands, while the Increasing Social job resources dimension had a lower level. These scores show a greater inclination of the sample to focus on concrete aspects and on the tough but challenging demands of voluntary work activity, but less commitment to the relational and social aspects therein.

As for OCB, the sample has high and similar scores for all the dimensions. It is interesting to note that Courtesy and Sportsmanship, two dimensions entailing a positive approach to activities to be carried out and to the relationships with fellow association workers, had slightly higher scores in spite of the potential for conflict or adversity.

As regards the MANOVA, all three social/personal variables considered are statistically significant: gender (Pillai’s trace = 0.01, F(3.108) = 5.09, *p* = 0.001), seniority (Pillai’s trace = 0.01, F(6.2418) = 3.04, *p* = 0.005), work role in the association (Pillai’s trace = 0.08, F(3.128) = 36.46, *p* < 0.001). The results of the univariate tests are summarized in Table 3.

There is a significant variation in the motivation to donate blood according to gender and the member’s role, a significant variation of JC according to the member’s role, and a significant variation of OCB depending on the seniority and the member’s role. The extent of the effects is in all cases extremely low and always very close to zero, indicating that these effects could in fact be irrelevant and attributable merely to the high availability of data.

### 3.3. The Relationship Between Motivation to Donate, Job Crafting and Organizational Citizenship Behavior

Table 4 summarizes the structure of the SEMs that have been adapted. The six adapted models were coded starting from 0, which identifies the baseline model. Each model indicates the covariance between Motivation to donate and JC and the relationship between the two factors Motivation to donate and JC, with the outcome JC variable being estimated or fixed at zero. Furthermore, we indicate whether or not the estimate has encountered convergence problems.

There were problems of convergence with the models codified with 0 and 1, in which the coefficients that describe the relationships between the Motivation to donate and JC and the variable OCB had been set to zero. Since these problems were generally a symptom of poor adaptation to the available data, these models were discarded. A new model was then adapted in which both the covariance between Motivation to donate and JC and the two coefficients that link Motivation to donate and JC to OCB are left free to vary (Model 2). Starting from this “full” model, the three parameters were individually constrained, looking for a better fit (models 3, 4 and 5). Model 4 still has convergence problems, though no difficulties were encountered with the estimation of models 3 and 5. When comparing the three estimated models, it was noted that model 3, in which both the covariance between Motivation to donate and JC and the relationship between Motivation to donate and OCB is estimated, was better than model 2 (ΔAIC = −2.00, ΔBIC = −7.00) and model 5 (ΔAIC = ΔBIC = −79.91).

Figure 2 shows the structure of model 3, together with the parameter values in the standardized solution. From a statistical point of view, the analyses highlight both the strengths and weaknesses of the tools, which inevitably affect the model proposed. Although RMSEA and SRMR have adequate values, CFI and NNFI tend to be problematic. In fact, the χ^2^ index model is significant (χ^2^ (2261) = 8799.65, *p* < 0.001), while both the RMSEA and SRMR have adequate values (RMSEA = 0.05, SRMR = 0.07). However, this particular model has comparative indices, equal to CFI = 0.80 and NNFI = 0.79, respectively.

CFI and NNFI are comparative indices, which evaluate the adaptation of the model with respect to a baseline in which the observed variables are considered to be unrelated. Therefore, the relationships between the observed variables may not be sufficiently high for the model architecture to be considered satisfactory. Given the previously declared exploratory and descriptive nature of the study, we decided not to alter the structure of the instruments adopted in order to preserve the descriptive ability of the constituent dimensions of the instruments themselves.

## 4. Discussion

CFA highlights the adequacy of the structure of the instruments used overall. This data is of interest because: (a) this study has confirmed—in regard to blood donation—previous research conducted on the Italian adaptation of the questionnaires used [48,55,56,58]; (b) the results can be a starting point for other studies, e.g., for the evaluation of the psychological variables considered, either specifically with regard to blood donation, or concerning voluntary activities in general.

Regarding the Motivation to donate, the most prominent are Values, Self-enhancement, Understanding, and Social motivation. On the other hand, Ego-protection and Career have lower values. This motivational profile is understandable if we consider that blood donation is defined in literature as pro-social, philanthropic and referable to social values such as gift, help, altruism [17,19,20]. For this reason, it is plausible to affirm that those who decide to donate blood, as well as being actively involved in the organizational activities for its collection, are driven by these motivational factors.

Ego-protection and Career motivations to donate can instead be defined as instrumental: the first to avoid negative or unpleasant experiences, the second because blood donation is seen as a means to achieve personal ends, instead of necessarily being philanthropic. For this reason, it is plausible that these motivations would have significantly lower scores.

JC also has high values overall. In particular, the Increasing structural job resources dimension denotes a tendency of volunteers to spend time on solving real and pragmatic problems, and the Increasing challenging job demands dimension refers to their inclination to face the most challenging demands of the organization’s activities. This sample profile is a plausible and interesting one, given that those who decide—in addition to donating blood—to actively engage in ensuring the success and proper functioning of this type of association, must be committed to solving real problems and facing complex challenges (e.g., managing and organizing other volunteers and donors, managing the collaboration with the many stakeholders interested in collecting and distributing blood, fund-raising etc. [3,49]).

Finally, the OCB scores are generally high and homogeneous. A profile of this type is more than plausible, considering that the propensity to engage in activities that are not prescribed, voluntary in nature, and often with a low level of structuring and formalization is typical of associations, and is essential for the proper functioning of this type of organization [41,42]. The slightly higher values of Courtesy and Sportsmanship would seem to suggest an affinity with relationships characterized by kindness, respect and attention for others (which also manifests itself in such associations); and a sense of belonging and identification with the organization, which leads volunteers to play down or disregard anything in their association they might disagree with or consider negative.

The MANOVA confirmed that although all the considered personal and social data variables have an influence on the psychological ones, the sample is not entirely irregular. Specifically, it showed that: females show a greater Motivation to donate than males; those with higher seniority (≥ 11 years) manifest a higher level of OCB; and those who hold a managerial role in an association show levels of Motivation to donate, levels of JC and OCB which are higher than in those who have an executive role. The first datum confirms previous results obtained using the same instrument in the Italian context [55,57]. The last two results are particularly interesting, considering how much they can affect organizational socialization and the consolidation of loyalty in volunteers, when nurtured by those with greater seniority and a higher role. For example, when knowhow and knowledge are shared, and their suggestions are advanced to implement innovative models of behavior and relationships that have proved to be valid in the light of their experience, concrete improvements ensue [10].

Regarding the relationship between the psychological constructs, considering each as a one-dimensional factor, the hypothesis that Motivation to donate and JC both predict the OCB, was only partially confirmed. The relationship that these three constructs have seems to be more structurally complex. In fact, a (positive) correlation emerges between the Motivation to donate and JC and, more importantly, JC has a clearly positive effect on OCB (hypothesis confirmed). Although our first piece of data will require further research to corroborate, it does seem convincing when we consider that the sample consists exclusively of unpaid volunteers, who are sensitive to the theme of blood donation and who have decided to actively engage in voluntary activities for its collection, and therefore to commit themselves to the work that is necessary to achieve this.

Our second piece of data upholds results of previous studies that have confirmed the positive influence of JC on OCB [50,51,52]. Furthermore, when considered together with the non-influence of Motivation to donate on the OCB (unconfirmed hypothesis), it is interesting to note that being motivated to donate blood is probably not enough to ensure a level of engagement in OCB that will enable the association to flourish and evolve. Indeed, what seems essential is that there is a propensity for JC to guarantee the identification (and sometimes the real creation) and realization of all the activities that promote and make blood collection possible and effective.

### Limitations and Future Research

There have been certain limitations in this study. The first is of a theoretical nature: the scarcity of the scientific literature explicitly linked to the three psychological constructs in question for the considered sample, did not allow the researchers to compare their results with similar studies. In fact, the theoretical precept on which the study is based (i.e., the JD-R Model [47,54]), can only be recognized in a general sense. This meant that the researchers could only ascribe an eminently explorative character to the study.

The second relates to the sample itself. Although our sample aimed to incorporate a series of socio-demographic variables and was quantitatively sufficiently large with regard to the reference population of AVIS volunteers [4], it relied on the availability of respondents to participate. It was also made up exclusively of AVIS volunteers, and although AVIS is by far the most important non-profit association in Italy for blood collection, it is not the only one.

In addition, the application of a cross-sectional and self-reporting methodology to collect the data may have affected the quality of measurements. The outcomes obtained need to be confirmed through future longitudinal research and qualitative studies.

Given that the collection and distribution of blood in Italy within voluntary associations is managed by both unpaid voluntary workers and salaried workers, one possible future area of research will be to assess and compare the different levels of JC and OCB between these two categories. Indeed, the ability of these types of organizations to provide an efficient blood collection and distribution service will be determined not just by the donors, but by the resulting synergy between the volunteers and salaried workers themselves.

## 5. Conclusions

In addition to the already abundantly studied motivational factors in blood donation, this study has highlighted the expediency and opportunity to also take into consideration the impacts of organizational commitment on the activities of these types of non-profit associations. This is particularly important for countries such as Italy, where the collection of such an essential good relies almost exclusively on the spontaneous, altruistic, and disinterested activity of unpaid volunteers, who not only donate their blood, but who freely commit their time and effort for its collection, processing, and distribution. This is indeed one of the association’s valuable but intangible psychological assets, which is proving to be of particularly fundamental importance in periods of crisis, which countries like Greece and Italy have been living through in the early years of this century. It has been a time in which the state has struggled to meet the needs of its citizens, and has necessitated the involvement of the voluntary sector, not only to encourage giving (in this case blood) but also the general pulling together and voluntary collaboration among ordinary citizens [8].

### Contribution to Knowledge and Practical Implications

From a practical point of view, this study has shown that philanthropic behavior—which in addition to the Motivation to donate, is well represented by JC and OCB—should not be solely the result of spontaneous cultural and social processes that are “external” to this type of association. For example, Boenigk et al. [10] have suggested a number of ways to consolidate and build a sense of belonging among donors in voluntary organizations. These include group activities, the development of online communities through social media, events dedicated not only to donors but also to volunteers, as well as other long-term strategies that shape the role of volunteers as co-creators of social values. Strategies designed to engage and build loyalty among volunteers by means of exploiting the experience of members with greater seniority and in higher positions, can be used to promote and maintain a sustained level of socialization among donors who decide to spontaneously engage in the organizational activities of the association.

In conclusion, this is an asset that needs to be enhanced, refined, and passed on to the next generation of volunteers, for the good of these types of non-profit associations and for the community itself, which will derive crucial benefits from their work.

## Figures and Tables

**Figure 1 ijerph-17-00934-f001:**
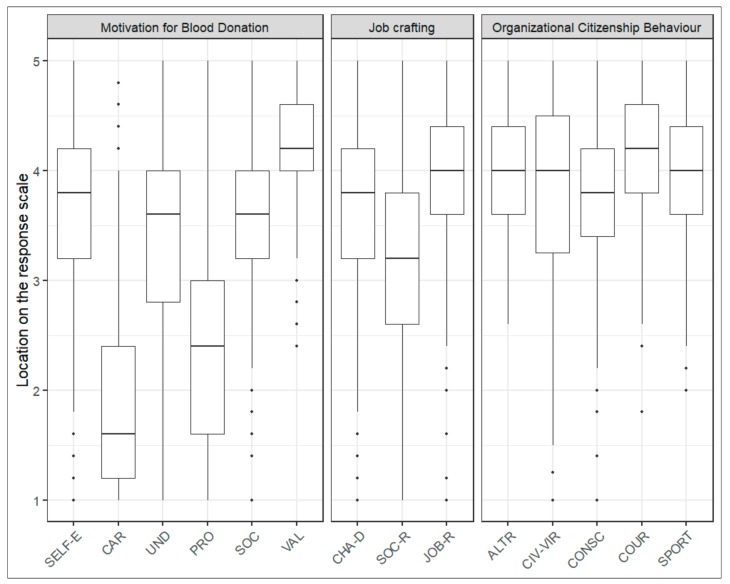
Location of the quantiles of the VFI Scale, JC Scale and OCB Scale on the continuum of the response scale.

**Figure 2 ijerph-17-00934-f002:**
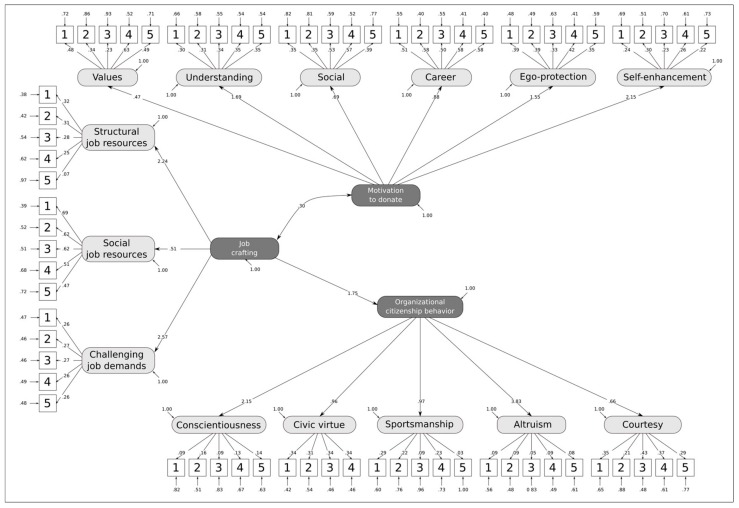
Path diagram and standardized parameters of the selected model.

**Table 1 ijerph-17-00934-t001:** Descriptive statistics of the total scale scores of the three VFI, OCB and JC Scales.

Scale	Alpha	Mean (SD)
Motivation for blood donation (VFI)		
Social (SOC)	0.63	3.57 (0.62)
Values (VAL)	0.50	4.25 (0.47)
Ego-Protection (EGO-P)	0.82	2.37 (0.90)
Self-enhancement (SELF- E)	0.71	3.68 (0.78)
Career (CAR)	0.84	1.83 (0.80)
Understanding (UND)	0.78	3.38 (0.83)
Organizational citizenship behavior (OCB Scale)		
Altruism (ALTR)	0.76	3.96 (0.58)
Sportsmanship (SPORT)	0.55	4.02 (0.55)
Conscientiousness (CONSC)	0.68	3.79 (0.68)
Courtesy (COUR)	0.66	4.10 (0.54)
Civic virtue (CIV-VIR)	0.82	3.87 (0.77)
Job crafting (JC Scale)		
Structural job resources (JOB-R)	0.69	3.95 (0.56)
Social job resources (SOC-R)	0.78	3.21 (0.78)
Challenging job demands (CHA-D)	0.85	3.70 (0.70)

**Table 2 ijerph-17-00934-t002:** Indices of adaptation of the three instruments.

Scale	Chi-Squared	RMSEA	SRMR	CFI	NNFI
VFI	χ^2^(390) = 1979.93, *p* < 0.001	0.06	0.06	0.87	0.85
OCB Scale	χ^2^(242) = 2260.28, *p* < 0.001	0.08	0.07	0.80	0.78
JC Scale	χ^2^(87) = 886.41, *p* < 0.001	0.09	0.07	0.89	0.87

**Table 3 ijerph-17-00934-t003:** Influence of the variables Gender, Seniority, and Member’s role on the construct scores.

		Gender	Seniority ^1^	Role (Manag. Vs. Exec.)
Motivation to donate	One-way Test	F(1,1210) = 13.21 *p* < 0.001, η^2^ = 0.01	F(2,1210) = 0.49 *p* = 0.612, η^2^ < 0.01	F(1,1210) = 4.32 *p* = 0.038, η^2^ < 0.01
	Post-hoc tests sig.		6–10 years vs. 0–5 years: *p* = 0.697 ≥11 years vs. 0–5 years: *p* = 0.082 ≥11 years vs. 6–10 years: *p* = 0.002	
	Means and SDs	Males: −0.07 (0.93) Females: 0.12 (0.90)	0–5 years: −0.04 (0.96) 6–10 years: 0.02 (0.96) ≥11 years: 0.01 (0.89)	Managers: 0.04 (0.91) Executives: −0.09 (0.94)
Job Crafting	One-way Test	F(1,1210) = 0.01 *p* = 0.914, η^2^ < 0.01	F(2,1210) = 2.77 *p* = 0.063, η^2^ < 0.01	F(1,1210) = 106.73 *p* < 0.001, η^2^ = 0.08
	Post-hoc tests sig.	Males: 0.02 (0.92) Females: −0.03 (0.97)	0–5 years: −0.20 (1.02) 6–10 years: −0.04 (0.95) ≥11 years: 0.09 (0.89)	Managers: 0.19 (0.86) Executives: −0.43 (0.97)
	Means and SDs		6–10 years vs. 0–5 years: *p* = 0.329 ≥11 years vs. 0–5 years: *p* < 0.001 ≥11 years vs. 6–10 years: *p* < 0.001	
Organizational citizenship behavior	One-way Test	F(1,1210) = 0.04 *p* = 0.844, η^2^ < 0.01	F(2,1210) = 4.19 *p* = 0.015, η^2^ = 0.01	F(1,1210) = 96.46 *p* < 0.001, η^2^ = 0.07
	Means and SDs	Males: 0.04 (1.89) Females: −0.07 (1.99)	0–5 years: −0.49 (2.03) 6–10 years: 0.05 (1.89) ≥11 years: 0.18 (1.87)	Managers: 0.37 (1.80) Executives: −0.85 (1.94)
	Post-hoc tests sig.		6–10 years vs. 0–5 years: *p* = 0.899 ≥11 years vs. 0–5 years: *p* = 0.158 ≥11 years vs. 6–10 years: *p* = 0.703	

^1^ For each dependent variable, a linear model was fitted and F-test results are reported. In the case of Seniority, the p-values of the three multiple comparisons (0–5 years, 6–10 years, ≥11 years), carried out by Tukey post-hoc test, are also shown.

**Table 4 ijerph-17-00934-t004:** Structure of the adapted models.

Model	Covariance between Mot and JC	Mot Influencing Variable of di OCB	JC Influencing Variable of OCB	Converged? ^1^
0	Fixed to 0	Fixed to 0	Fixed to 0	No
1	Free	Fixed to 0	Fixed to 0	No
2	Free	Free	Free	Yes
3	Free	Fixed to 0	Free	Yes
4	Free	Free	Fixed to 0	No
5	Fixed to 0	Free	Free	Yes

^1^ Yes = Converged; No = Not converged.

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
