# Peer review of "Motivation to Donate, Job Crafting, and Organizational Citizenship Behavior in Blood Collection Volunteers in Non-Profit Organizations"

_ijerph, 2020, doi:10.3390/ijerph17030934_

Round 1

Reviewer 1 Report

This is an interesting study dealing with a very important topic . The technical side of the paper is generally well handled. However, the literature review of the various concepts utilized in the study needs extension, particularly of organizational citizenship behavior of which there is a large and growing body of research. The comment in line 180 of the manuscript : "Scientific studies that have focussed on the three constructs.....are scant" , is incorrect as there is a large extant literature in each case. It is true, however, that bringing the concepts together as independent variables in the explanation of the behavior of blood organization volunteers in Italy is a novel contribution to the literature.

 A  problem in the research design is the use of the concepts of organizational citizenship behavior and job crafting as they relate to volunteers who are not formally employees of blood donor organizations. for example, the concept of organizational citizenship behavior has real relevance only in situations where employees/members of an organization are not voluntary workers and hold remunerated positions.This mismatch becomes evident in the various sub constructs of the scales themselves despite the orientation towards "association" in the items themselves. The authors need to discuss this aspect in more depth by showing how the relevance of the concepts with regard to voluntary associates rather than formal employees may have a bearing on the findings. The authors are aware of the possible difference between 'ordinary donors' and voluntary  associate donors affiliated with a blood collection organization, but this aspect is not developed suffiently in the discussion. Clearly also the concepts of job crafting and organizational citizenship behavior would not be relevant in the case of ordinary donors and their application would then presumably not be possible in future research studies.

Reviewer 2 Report

I would like to congratulate the authors for their revision. I thoroughly read through the revised manuscript and checked for the many changes that the authors had described very comprehensibly in their point-by-point response. There are no further requests from my part. I recommend to accept the manuscript for publication.

Round 2

Reviewer 1 Report

I have read your explanations and  the substantive changes that you have made to the paper and believe your article should be accepted for publication once minor spelling ( e.g. volonteers rather than volunteers) and grammar errors have been corrected. 

Author Response

This manuscript is a resubmission of an earlier submission. The following is a list of the peer review reports and author responses from that submission.

Round 1

Reviewer 1 Report

The paper “Motivation to donate, Job crafting and Organizational citizenship behavior in blood collection volunteers in non-profit organizations” aims to describe motivation for blood donation, propensity to Job Crafting (dimension of orientation to improvement and innovation) and Organizational Citizenship Behavior (dimension of commitment in non-prescribed and extra-role activities) as possible key factors to keep a proper management of the association.

Authors propose an original position by putting together the gift of blood, job crafting and OCB this kind of contribution on a topic still poor investigated in psychology deserves attention.

Below you’ll find some remarks on the paper.

Introduction

Better explain the reasons for choosing these constructs to support the motivation of voluntary donors.

Measure

Explain the reasons that led to adapt the JC Scale and the OCB Scale to the volunteering context. Why did the authors not explore their presence in donors even outside of voluntary organizations?

Analysis

It seems that for the analysis authors put together different roles (donors engaged as volunteers and donors engaged as presidents, etc.): perhaps they could do the analysis on the two distinct groups. Or maybe they could also do an analysis related to the duration of the “career as donors”.

Discussion

From line 456 to line 469 – widen and deepen the possible hypotheses on the relationship between the 3 constructs

In sum, I think the paper can be published with minor revision, but authors should explain better

explain the theoretical assumptions of their study.

Reviewer 2 Report

This is an interesting study that fits well with the special edition's topic and scope. The paper focuses on motivation for blood donation, propensity for improvement /innovation, job crafting and organizational citizenship behavior. The authors analyse the data with scientific  thoroughness and indicate problems and shortcomings in the statistical models generated and the internal consistency of some of the items in the scales used, some of which have particularly low values.The sample is large however deals only with the attitudes of blood donors who are already voluntary members of Italian blood donor associations. In this regard It is not clear in the biographical  description of the sample, who was employed in gainful work activity and those who were full time voluntary members of a blood association. For instance we are told that 69.7% of the sample were workers and that most ( 69%) had managerial roles in the association , with 30.6% holding executive roles. This seems to suggest  that over half ( 69%) were in full-time gainful employment,but did voluntary work for the association as well. This aspect needs clarification as it is of particular importance to one of the core measurements,that of job crafting, What would be a special contribution would be to note differences between sample respondents' scores on the measuring instruments and their employment status. clearly those who work as well as perform voluntary association duties may have distinct profiles and these profiles could be specially important in finding future blood associates. This aspect needs further discussion in the paper as do the practical implications of the study which could spring from an analysis of this kind.

The paper is generally well-written, but Figure 1, 2 and 3 are difficult to read and need further attention       

Reviewer 3 Report

The manuscript focuses on a relevant social issue, the blood donor process, and explores new possible factors (job crafting) and outcome (organizational citizenship behavior). I enjoyed reading this paper, nevertheless there are some parts of the paper that can be improved.

1) Subjects of the research have a double role, being both donors and members of an organization. I believe that this needs to be clarified at the beginning, also providing some more information about the Avis organization; otherwise it’s hard understanding why authors purpose to study motivation to donate together with variables related to organizational roles. For instance explaining how many employees Avis has, or if the participants are employees or volunteers (and, if so, which is their contribute). Further, it’s not clear to me how most of them, that is about 893 participants, can have a managerial role (see p. 5, line 204). I think that this is important because the proposed research model is strictly related to this system of blood collection, were an association mediates between citizens and health services.

In line with this, results could be discussed also for their practical implications (e.g., how implementing policies aimed at supporting not only the motivation of single donors but also their commitment in supporting the blood collection process).

2) The theoretical sections (1.2, 1.3) should describe how the JC and the OCB constructs are relevant for the blood donor or they are related each other, consistently with the proposed model. I realize that the literature is still spare and you can’t draw on sound previous research, anyhow it’s important to provide some conceptual support to your study, for instance explaining how the two constructs can be declined in the associative work contexts, or are specifically related to the blood donor process, or also considering them within a wider theoretical framework (e.g. Blau’s social exchange theory, or Bandura’s agentic perspective).

As well, Authors’ hypotheses need to be clearly stated.

I also suggest moving the description of the measurement instruments in the Methods section.

3) Results are really analytic. I appreciated the description of all preliminary tests you did, anyhow I find they go beyond the purpose of the paper, so I suggest to synthesize them. For instance figures 1, 2 and 3 can be erased. Also the analyses of the role of socio-demographic variables is too detailed related to the paper’s aims. Alternatively, you can include the psychometric validation of the scales in the aims, or further hypotheses related to the gender–seniority–role contribute to the donor process.

4) Some sentences need to be clarified:

–“Therefore, JC represents an individual strategy to promote changes, innovations and challenges that can generate a work and organizational improvement.” (p.4) 

– “Respondents were not identified by random extraction methods” (p.5)

Reviewer 4 Report

The manuscript describes the hypothesis, study, and methods of an original research. The Authors aimed to assesses the level and describes the relationship between motivation to donate, job crafting propensity, and the organizational citizenship behavior of blood collection volunteers in a non-profit association. It is a current and interesting theme and there is a need to study this issue in the workplace, especially among blood donors.

The language and the grammar require a proof-reading in order to correct several inaccuracies and to improve reading fluency. Abstract and keywords are enough both in terms of appropriateness of context and the purpose of study.

The introduction is sufficiently written and satisfactory in terms of appropriateness of context and the purpose of study. The analysis of the literature can be improved.

Since the second half of the 2000s, a global economic crisis - whose effects are still ongoing in some countries like Italy - has occurred. Therefore, you could also consider a brief discussion regarding the potential additional effects of the economic crisis on the dynamics of work-related stress and their impact on health. I state that there are no specific references to literature, and, especially for this, it would be interesting that you reflect on this aspect. For example, can a global economic crisis interfere with motivation to donate? With regard to the dynamics of the economic crisis, you can refer to the following publications:

Mucci N et al. The correlation between stress and economic crisis: a systematic review. Neuropsychiatr Dis Treat 2016; 12:983-993. doi: 10.2147/NDT.S98525. Sotiropoulos DA and Bourikos D. Economic crisis, social solidarity and the voluntary sector in Greece. Journal of Power, Politics & Governance2 (2014): 33-53.

Methods section appears of a more than sufficient quality. Add also any possible consideration about ethics statement in the Methods body text. The statistical methodologies that you have used are sufficiently well illustrated.

Results section is, as a whole, of a satisfactory quality. However, you should better connect the text with the 4 tables and 5 figures. In other words, you should more thoroughly explain the contents of the tables and the figures before referring to them in the body text.

The discussion section is overall sufficient. You should more carefully compare your findings with the field literature. I suggest you go deeper into the study's limitations. Finally, I suggest you use a final paragraph to sum up your findings. Otherwise, you should also carefully explain what is the specific contribution that your findings bring to literature and knowledge in this area.

Check accurately all the quotes in the brackets in the text and in the Reference section. These must strictly comply with the Author's guidelines.
